# Animal-Assisted Interventions for the Improvement of Mental Health Outcomes in Higher Education Students: A Systematic Review of Randomised Controlled Trials

**DOI:** 10.3390/ijerph182010768

**Published:** 2021-10-14

**Authors:** Charlotte Parbery-Clark, Marvellas Lubamba, Louise Tanner, Elaine McColl

**Affiliations:** 1Population Health Sciences Institute, Newcastle University, Newcastle upon Tyne NE1 7RU, UK; lubamba@ualberta.ca (M.L.); louise.tanner@newcastle.ac.uk (L.T.); elaine.mccoll@newcastle.ac.uk (E.M.); 2Newcastle City Council, Civic Centre, Newcastle upon Tyne NE1 8QH, UK

**Keywords:** animal-assisted interventions, mental health outcomes, stress, anxiety, higher education, systematic review

## Abstract

Background: The aim of this systematic review was to evaluate the effectiveness of Animal-Assisted Interventions (AAIs), particularly Animal-Assisted Therapy (AAT) and Animal-Assisted Activity (AAA), in improving mental health outcomes for students in higher education. The number of students in higher education reporting mental health problems and seeking support from universities’ student support services has risen over recent years. Therefore, providing engaging interventions, such as AAIs, that are accessible to large groups of students are attractive. Methods: MEDLINE, PsycINFO, Embase and Cochrane Library were searched from relative inception to end of April 2020. Additionally, a grey literature search was undertaken. Independent screening, data extraction and risk of bias assessment were completed, with varying percentages, by two reviewers. Results: After de-duplication, 6248 articles were identified of which 11 studies were included in the narrative synthesis. The evidence from randomised controlled trials suggests that AAIs could provide short-term beneficial results for anxiety in students attending higher education but with limited evidence for stress, and inconclusive evidence for depression, well-being and mood. For the non-statistically significant results, the studies either did not include a power calculation or were under-powered. Conclusions: Potential emerging evidence for the short-term benefits of AAI for anxiety, and possibly stress, for students in higher education was found.

## 1. Introduction

Attending higher education commonly represents a major life transition for young people, with it often being the first time living away from the family home, which can bring social, financial and academic stressors [1,2]. The true prevalence of mental health problems for students in higher education is hard to estimate accurately. For example, in the UK, there is a scarcity of large-scale studies being truly representative of the UK student population in higher education or applying a weighted adjustment to accommodate for the lack of representativity [3,4]. However, the number of students in higher education disclosing mental health problems and accessing higher education institutions’ (HEI) support services has risen in recent years [1]. Disclosure and requesting support can result in long waiting lists for more traditional individualised therapy sessions, while stigma around seeking help for mental health and well-being is still present [5,6]. Therefore, a possible solution may be the provision of interventions aimed at reducing stress and anxiety as well as boosting mental health and well-being that are appealing, effective, and accessible to large groups of students [6]. In this respect, part of the solution could be Animal Assisted Interventions (AAIs). 

AAI is an umbrella term that describes the use of various animal species in numerous ways that are beneficial to humans, and includes Animal-Assisted Therapy (AAT), Animal-Assisted Education (AAE), Animal-Assisted Activity (AAA) and more recently, Animal-Assisted Coaching (AAC) [7,8,9]. In summary, AAT is a structured and goal-directed intervention with a specifically trained live animal and is designed to ameliorate socio-emotional, behavioural, cognitive and/or physical functioning [7,8]. AAA is a planned informal interaction with human-animal teams for recreational, motivational and educational opportunities [7,8]. AAE, similarly structured to AAT, focuses on specific educational or academic goals with a professional trained in, and with expertise, in education or a similar field [7,8]. AAC is also similarly structured to AAT and AAE but delivered by licensed coaches focusing on personal growth [8]. AAIs’ popularity have risen over recent years, and they are used in diverse settings such as hospitals, nursing homes, schools and universities [10,11,12,13,14]. Positive interactions with animals have been shown to have beneficial human physiological responses, such as reduction in heart rate, blood pressure, stress hormones (for example, cortisol), and increase in hormones associated with positive emotions (for example, oxytocin) [15,16,17,18]. Evidence has emerged that AAIs, particularly AAT, may be effective in treating various mental health conditions (such as schizophrenia, depression and drug/alcohol addiction), developmental disorders (such as autism-spectrum disorder) and depressive symptoms in individuals with certain neurological conditions (such as dementia) [10,11,14,19]. Furthermore, a recent meta-analysis, involving both children and adults, demonstrated statistically significant improvements in heart rate, self-reported anxiety and stress, but not blood pressure, after AAIs [20]. The discrepancies in findings related to BP may be due to psychological arousal and do not necessarily contradict the stress relieving effects [21]. 

There are numerous theories of why and how AAIs may work. For example, Crossman et al. [6] suggests various theories for how animals may reduce stress including: emotional contagion (transmitting the animal’s positive emotions onto humans)facilitating social interactionopportunities for reinforcement (by partaking in pleasurable activities and experiencing positive emotions)evoking expectations that participation will reduce stress (expectancy that the intervention will work)

Beck [22] describes that the human–animal bond is rooted in evolutionary as well as physiological and psychological processes with significant health benefits for both humans and animals. Furthermore, the importance, in the psychosocial model, of social support for health and how social support can function as a buffer against stress are also relevant [23,24]. The animal–human bond can be considered as a type of social relationship, which can offer this type of support. Some individuals may form an animal–human bond more readily than a human-human bond as animals are considered to be indifferent and non-judgemental to an individual’s appearance, social skills or socioeconomic status (SES) [25]. Furthermore, the Biophilia Theory proposes that humans are drawn to interact with animals due to an innate desire to connect with living organisms and nature [25,26]. Additionally, distraction as a cognitive refocus may also contribute, though this research has mostly focused on anxiety and pain whilst awaiting or receiving medical treatment [27]. 

Interestingly, the prevalence of programmes using AAIs at HEIs has increased, for example by 2015 over 900 existed in the USA [6]. In the UK, these types of programmes have also risen in popularity with various forms being implemented, ranging from one-day events to specific sessions [28,29,30,31,32]. Additionally, the evidence-base for using AAIs with this population is growing with persuasive descriptive and anecdotal reports of the benefits [2,6,33,34]. Over recent years, more randomised controlled trials (RCTs) have been published, evaluating the effectiveness of AAIs in respect of various outcomes for students in higher education [35,36,37]. Nonetheless, a lack of completed systematic reviews for AAIs and this specific population exists. Therefore, to our knowledge, no completed systematic reviews were identified that primarily evaluate the effectiveness of AAIs, particularly AAT and AAA, delivered in multiple or single sessions, in improving mental health outcomes for students in higher education, with no age or course restrictions. This systematic review addresses this gap in the literature to help inform HEI providers regarding the potential benefits of AAIs for students’ mental health and well-being, as well as to provide recommendations for future policy, practices and research.

## 2. Aim and Objectives 

The aim of this systematic review was to evaluate the effectiveness of AAIs, particularly AAT and AAA, in improving mental health outcomes for students in higher education. 

The objectives were to: systematically search and critically appraise the relevant published and unpublished literature on the effectiveness of AAIs, particularly AAT and AAA, in improving mental health outcomes for this particular population.provide evidenced-based recommendations for policy, practice and further research.

## 3. Methods

### 3.1. Protocol and Ethics

A scoping review defined the focus of this systematic review by identifying gaps in the literature. This included searches for published literature in MEDLINE, The Cochrane Library, PsychINFO and Campbell Collaboration, as well as PROSPERO and Joanna Briggs Institute’s Systematic Review Register. The protocol for this systematic review was peer-reviewed and registered on the PROSPERO database on 25 June 2020 (registration number CRD42020186541) [38]. The Preferred Reporting Items for Systematic Reviews and Meta-Analyses (PRISMA) guidelines were followed [39]. Ethics approval was not required. 

### 3.2. Search Strategy

The search strategy was independently peer-reviewed by both an information specialist and an experienced librarian at Newcastle University. The full search strategies are included in Appendix A. The search strategies were not limited by year, study design, language or publication status. MEDLINE, PsycINFO, Embase and Cochrane Library with inclusion of Central Register of Controlled Trials CENTRAL, Cochrane Database of Systematic Reviews and Cochrane Clinical Answers were searched from their relative inception to week three of April/week commencing 27 April 2020. Additionally, an Advanced Google search, using the first four pages due to Google sorting by relevance, during the week ending the 1 May 2020, as well as a further search in the PROSPERO database were completed. To identify additional studies, the reference lists of all full manuscripts meeting eligibility criteria and a “cited in” search using Science Citation Index/Science Citation Index Expanded via Web of Science were reviewed. 

### 3.3. Inclusion and Exclusion Criteria 

A description of the inclusion and exclusion criteria, according to PICOS (Population, Intervention, Comparator, Outcome and Study design), has been provided below and summarised in Table 1 [40]. During study selection, no restrictions for study geographical location, date or language were applied. 

The population was students in higher education with no age, course or location restrictions. Higher education was operationalised as delivered beyond secondary school leading to a degree [46]. Studies that only included students with an already established diagnosis were excluded, as this would have substantially affected the clinical heterogeneity of the studies being compared. Consequently, the intervention’s true effect might have been affected by differences in the population and not the intervention itself, thereby potentially compromising the generalisability of the results [40].

Differences regarding the definitions, corresponding terminology and operating practices of the various types of AAIs can lead to difficulties assessing and comparing the interventions [7,8,9,47,48,49]. AAA and AAT are often used interchangeably in the literature, leading to ambiguity [12,49]. To overcome these identified discrepancies, both AAT and AAA were included. The definitions provided by the International Association of Human-Animal Interaction Organisations (IAHAIO) and the American Veterinary Medical Association (AVMA) to classify the various types of AAI were used [7,8]. In summary, AAT involves a specifically trained live animal in a planned, structured and goal-directed intervention, designed to improve socio-emotional, physical, behavioural and/or cognitive functioning of the individual(s) as part of the treatment process [7,8]. AAT is delivered and/or directed by a trained human professional (from education, health or human services) with specific expertise, and progress is measured/evaluated [7,8]. AAA is a planned informal interaction with trained animal-human teams for recreational, motivational and educational opportunities [7,8]. For the purposes of this systematic review, in accordance with IAHAIO, AAA is goal-orientated [8]. It was anticipated that studies might provide insufficient detail to objectively assess and classify the type of intervention (for example to distinguish clearly between AAT and AAA). Consequently, studies were included if the live animal was called/considered a therapy animal, a therapeutic aim/goal was identified, and the outcomes of interests were evaluated. If the term “therapy animal” had not been used, the authors had to explicitly mention that the animal had, at least, had introductory training and an assessment/evaluation [8,9]. If the intervention was part of a multi-component programme, isolating the effectiveness of the AAT/AAA had to be possible; otherwise, the study was excluded. Additionally, if a study used a stressor, the stressor had to be an aspect of study, training, education or be student specific. An exam or an experimental cognitive test that was used to emulate evaluative testing are examples of included stressors.

Any type of comparator was included, including active intervention, attention control, placebo/sham therapy, usual care/treatment as usual, or alternative active intervention. If there was more than one comparator, one was chosen according to a hierarchy that was established to assess the intervention’s effectiveness [40,50]:control (no-treatment, attention, usual care, or wait-list)validated sham treatment (where known not to be efficacious)other active intervention with known efficacysham/alternative treatments (where efficacy is unknown)

Specific psychological mental health outcomes, assessed using various established or published standardised measures, before and after the intervention were included. The primary outcomes of effect focused on anxiety and/or stress, using a range of established or published standardised measures, including but not limited to, Perceived Stress Scale (PSS) or State-Trait Anxiety Inventory (STAI) [51,52]. These outcomes were chosen as students in higher education may experience a significant amount of stress [6,53,54,55]. Additionally, anxiety can occur as a reaction to stress, with stress and anxiety being closely linked [56]. Differences in anxiety and/or stress scores from baseline pre-intervention to directly post-intervention and/or final follow-up were included. Secondary outcomes of effect focused on mood/affect, depression and well-being, using a range of published or established standardised measures, including but not limited to, Warwick–Edinburgh Mental Well-being Scale (WEMWBS) or Positive and Negative Affect Schedule (PANAS) [57,58]. The time-point considered to have the biggest potential health benefit was considered as the time-point immediately after the intervention. Thereafter, the next best alternative was the time-point closest to the end of the intervention.

RCTs represent the gold standard for measuring an intervention’s effectiveness with high internal validity [59,60]. Therefore, only RCTs were included. Studies were excluded if allocation to the respective groups was not objectively randomised, for example if randomisation was according to participants’ availability, student number or date of birth with no random component. Any pilot/exploratory studies that met all inclusion criteria and analysed the outcomes of interest were included, unless the full RCT had been reported.

### 3.4. Study Selection

Following the electronic database and grey literature searches, titles and abstracts (*n* = 8036) were de-duplicated. The remaining titles and abstracts (*n* = 6248) were cautiously screened for relevance, erring on over-inclusivity, by the first author (CPC) in Rayyan to remove obvious irrelevant studies or duplicated studies not identified by the automated systems [40,61]. Subsequently, 100% of the articles that were deemed potentially relevant (*n* = 928), were reviewed independently by two reviewers (CPC and ML, a fellow Master’s student) against the pre-specified eligibility criteria using Rayyan [61]. 132 articles were identified as requiring full manuscript review, which was undertaken in full by CPC and 20% by ML, both blinded to the other’s decisions. To identify a random 20% for ML, the titles were arranged alphabetically in Endnote. Subsequently, a random number (*n* = 114) was generated by a true random number generator website [62]. Every fifth article (as 20%) was chosen starting from the 114th article. If any discrepancies arose at any stage, discussion occurred between the two reviewers. If consensus was not achieved, agreement was obtained by discussing with a third reviewer (EM). If the full manuscripts were not available, title/abstract/keywords were reviewed. To meet the eligibility criteria to request an inter-library loan, at least three elements of the PICOS criteria had to be fulfilled. Keywords were identified from Rayyan, Endnote and MeSH analyser [61,63,64].

### 3.5. Data Extraction

A structured Microsoft Excel data extraction form was adapted with permission of Dr. C Marshall after being piloted with two studies. The TIDieR checklist was incorporated as AAI is a complex intervention [65]. The data extraction form included [13,66]:study characteristics (such as design, setting and country)participants (including eligibility criteria, age, gender, and type of student)interventions (for example single/multiple sessions, species of animal, if handler present, duration and frequency of sessions as well as length of programme)outcomes (such as the relevant measures used, interpretation, results and time-points for measurements)

Data extraction with rigorous double-checking was primarily undertaken by CPC. ML independently data extracted 18% (*n* = 2 out of 11). The same previously described strategy for addressing disagreements was followed, as required.

### 3.6. Risk of Bias Assessment and Strength of Evidence

A validated tool, Risk of Bias 2.0-revised (RoB2) for individually randomised parallel-group trials, was used for the risk of bias assessment [67]. As this systematic review was aimed to inform a health policy question, the effect of interest was the effect of assignment to AAIs [68]. For each study that reported more than one of the relevant primary or secondary outcomes, a risk of bias assessment was completed for each relevant outcome. For each study that reported multiple time-points for the assessment of outcomes or had more than one comparator, one time-point and one comparator were chosen according to the hierarchies previously described. Where the RoB2 guidance did not cover specific situations found in this review, decision rules were developed and applied in a standardised manner (Appendix B). The RoB2 assessment was completed in full by CPC with 18% (*n* = 2) undertaken independently by ML. The same previously described strategy for addressing disagreements was followed. Authors were contacted to request clarifications or to access missing data and given a two-week period to reply.

A narrative synthesis based on the Economic and Social Research Council (ESRC) Methods Programme was planned [69]. Meta-analysis was not appropriate due to all studies being at high risk of bias for the outcomes of interest, with most having high or some concerns regarding missing data, alongside the substantial clinical heterogeneity. Since meta-analysis was judged to be inappropriate, a harvest plot using vote counting, based on direction of effect, was used with categorisation of the studies by their effect (detrimental, no or beneficial effect) [40]. Effect size or statistical significance were not included for this categorisation as this can be misleading [40]. Vote counting was used for both the primary and secondary outcomes using mean change score for one comparator and one time-point according to the hierarchy previously described. If the outcomes were measured immediately after the intervention, as well as, at an additional time with a stressor, the former was only included for the vote counting. A set of decision criteria were created for interventions with a stressor and those without to standardise interpretation of the expected response to the intervention (Appendix B).

The quality and relevance of evidence was appraised using the ’Weight of Evidence’ approach [70] and described in the guidance on narrative synthesis for the ESRC guidance [69]. In accordance with the ESRC guidance, trustworthiness was measured by Jadad’s scale [69,71].

## 4. Results

### 4.1. Study Selection

Eleven articles, describing eleven studies, met the inclusion criteria. The PRISMA flowchart is displayed in Figure 1. Ten studies were journal articles and therefore classed as published literature [35,37,72,73,74,75,76,77,78,79]. One study was a dissertation for partial fulfilment of PhD and classed as grey literature [80]. All 11 studies were individually randomised parallel-group trials. Of these, three were described as pilot or exploratory studies but nonetheless presented data on interventions’ effectiveness [37,75,80]. The excluded full text published studies (with reasons) are listed in Appendix C.

### 4.2. Study Characteristics

Six studies were conducted in USA [72,75,76,77,79,80], three in Canada [35,73,78], one in Scotland [37], and one in Austria [74]. Sample sizes (for those randomised) across all included studies varied from 20 to 357 students.

#### 4.2.1. Population

For the majority (9 studies), the participants were university students [35,37,72,73,76,77,78,79,80]. Seven studies reported the characteristics according to the recruited sample size [35,37,72,73,76,77,80] and three for the analysed sample size [74,78,79]; one reported the characteristics of the students attending the degree programme from which the participants were recruited and not the sample’s characteristics (this study is excluded from the descriptive statistics below) [75].

Ten studies reported gender with the majority (ranging from 57% to 85%) of participants being female [35,37,72,73,74,76,77,78,79,80]. The most common age bracket was 20-year-olds and under, followed by 21 to 25-year-olds. Type of student was clearly reported in five studies [35,73,76,77,79]. Of those, 80% were undergraduates [35,73,76,77]. The students’ year of academic studies was fully reported in two studies [35,74]. Five studies reported ethnicity [35,73,76,77,80]. None of the studies reported health status or socio-economic status (SES). The participants studied a range of course subjects (such as psychology and nursing). Three studies collected the potential confounder of pet ownership [37,72,78], and one reported experience with the animals employed in the intervention (horses) [80].

#### 4.2.2. Intervention

Ten studies employed dogs [35,37,72,73,74,75,76,77,78,79] and one employed horses [80]. Group sessions were the most common format: seven studies implemented this session type [35,37,72,74,76,77,78], whilst two implemented individual sessions [73,80]. A further two studies implemented either a combination of group and individual sessions [75] or the format of sessions was not clearly reported [79]. The student to dog ratio in the group sessions was clearly reported in two studies [35,37] and implied in three [74,76,77], ranging from 3 to 4-students-per-dog to 12 to 14-students-per-dog.

The presence of a handler during the sessions was clearly reported in seven studies [35,37,72,74,78,79,80]. Nine studies allowed free interaction with the animals [35,37,72,73,75,76,77,78,79] and two used a structured format [74,80]. Most interventions (*n* = 8) corresponded to the definition of AAA described in 3.3 [37,72,73,75,76,77,78,79]. Two studies were classified as AAT [35,74] and one combined AAT and AAE together [80]. These classifications were mostly from an objective assessment of the description provided vis-à-vis the definitions outlined in 3.3. If insufficient detail for an objective assessment, the classification used by the primary authors was kept [35]. Length of intervention ranged from unspecified to 90 min, with the modal length being between 10 to 20 min. Seven studies used a single session [35,37,72,73,78,79,80]. Four studies used multiple sessions [74,75,76,77]; of these, two used once-weekly sessions for four weeks [76,77]; one implemented three sessions with non-reporting of the time interval between sessions [74]; and one allowed various lengths and frequencies over a 15 to 16-week period at the participant’s choice [75]. None of the studies reported monitoring or measuring the intervention’s fidelity. The theoretical frameworks were clearly stated in three studies [35,37,80]. More information regarding the theoretical frameworks is included in Appendix D.

#### 4.2.3. Outcomes

The primary outcomes were self-reported anxiety and stress measured before and after the intervention. Seven studies measured self-reported anxiety [37,74,75,76,77,79,80] with most using the State-Trait Anxiety Inventory (STAI), apart from one study which used the Hospital Anxiety and Depression Scale (HADS) [75]. Two studies measured stress using the Perceived Stress Scale (PSS) on the original 5-point Likert scale [35,78].

Regarding the secondary outcomes, two studies measured depression [75,77] using the HADS-depression subscale and Beck Depression Inventory II (BDI II), respectively. Five studies measured mood/affect, with most using Positive and Negative Affect Schedule (PANAS) [72,73,76,78] and one used the University of Wales Institute of Science and Technology Mood Adjective Checklist (UMACL) [37]. Well-being was measured in three studies [35,37,78] using various tools.

The timing of outcome measurement after the intervention varied substantially between studies (with some reporting multiple time-points outlined in Table 2) and encompassed: immediately after the intervention without a stressor (*n* = 6); after the intervention but before an exam (*n* = 2); after the intervention and an experimental stressor (*n* = 4); within 24 h of the intervention (*n* = 1); 2 weeks after the intervention (*n* = 1); up to 1 month after the intervention (*n* = 1); up to 15–16 weeks since the start of the intervention (*n* = 1).

Regarding adverse events reporting, Hall [75] described ten individuals who reported increased anxiety and stress due to being in the control group and unable to participate with the animal. These participants were subsequently removed from the study to allow interaction. No other adverse events were reported. None of the studies specifically reported any adverse events for the animals involved.

Characteristics are summarised in Table 2.

### 4.3. Risk of Bias Assessment

Figure 2 summarises the risk of bias for each domain with an overall assessment for each study for all the relevant outcomes. All studies were at overall high risk of bias for the outcomes of interest as susceptible to high risk of bias in the domain “measurement of the outcome”. This was mainly due to the nature of the intervention as participants could not be blinded and the measures were self-reported, therefore were not objective.

### 4.4. Strength of Evidence 

The quality and relevance of evidence was appraised using the ’Weight of Evidence’ approach [70] and described in the guidance on narrative synthesis for the ESRC guidance [69]. The strength of evidence of the included studies is summarised in Table 3. One study had high overall weight [74], eight [35,37,73,75,76,78,79,80] had medium overall weight and two had low overall weight [72,77].

### 4.5. Narrative Synthesis: Interventions’ Effect

As all the studies were at high risk of bias for the outcomes of interest, with most having high or some concerns regarding missing data, alongside the substantial clinical heterogeneity, a meta-analysis was not appropriate as it was unlikely to provide meaningful results.

All the included measurement scales provided continuous data; results were presented in various formats such as pre- and post-values, mean change or adjusted estimates of the intervention’s effects (for example, using analysis of covariance (ANCOVA) with baseline measurements included as a covariate). Appendix D provides a summary of the studies’ results with mean change (post-scores minus pre-scores) as the common parameter using the most immediate measurement of outcomes after the intervention (or the next best alternative).

#### 4.5.1. Primary Outcomes: Anxiety

Seven studies reported the intervention’s effect on self-reported anxiety levels [37,74,75,76,77,79,80]. Six studies employed dogs [37,74,75,76,77,79] and one employed horses [80]. Four studies used group sessions [37,74,76,77], one used individual sessions [80], one offered both group and individual sessions [75] and one did not explicitly state the type of sessions but was inferred to be individual [79]. Session length varied and included student’s choice to sessions lasting 12, 20 and up to 60 min. Three studies offered a single session [37,79,80], two offered four sessions (once/week) [76,77], one varied depending on students’ choice [75] and one offered three consecutive sessions but did not report the time interval between sessions [74]. Most interventions were consistent with AAA [37,75,76,77,79] and of these studies (*n* = 5), two included, alongside the dogs, interactive games, icebreakers and snacks for the participants/dogs [76,77]. Furthermore, Gebhart et al. [74] offered AAT and Meola [80] offered tailored and structured individual sessions called equine-assisted learning supervision (EALS). The latter incorporated both educational and therapeutic goals, therefore combining AAE and AAT.

Six studies tested outcomes without a stressor, with four showing a statistically significant improvement in favour of the AAI [37,74,75,77]. Four studies tested the outcomes immediately after the intervention without a stressor, with three showing a statistically significant reduction in favour of the AAI [37,74,77] and one showing a non-statistically significant reduction but did not include a power calculation [76]. For the two studies which tested the outcome after a longer time interval, Meola [80] found a non-statistically significant reduction in anxiety when measured up to one month after the intervention but this study was under-powered. Hall [75] found a statistically significant reduction in the post-intervention scores 15–16 weeks after the intervention had started.

Four studies involved a stressor: two assessed the outcomes after the intervention but prior to an exam [74,79] and two assessed the outcomes after an experimental cognitive test (WAIS-IV IQ test) which was designed to imitate evaluative testing that occurs in higher education [76,77]. Two studies found no statistical difference [74,77] but a power calculation was not included. Hunt et al. [76], despite no significant effect of condition on anxiety after a stressor, still undertook paired comparisons which showed a statistically significant higher level of anxiety in the AAI group compared to the control group. Williams et al. [79] showed an increase in anxiety for both groups, after the intervention but before an exam, which one might therefore expect. However, the control group had statistically higher anxiety levels than the AAI group.

Overall risk of bias was high for all seven studies regarding the outcomes of interest. Regarding overall strength of evidence, one study had high [74], one had low [77] and the remaining five had medium strength. Using vote counting, according to direction of effect and not statistical significance as described in the methods, all seven studies showed a beneficial effect in favour of the intervention compared to the comparator (demonstrated in harvest plot in Figure 3).

#### 4.5.2. Primary Outcome: Stress

Two studies, using single group sessions with dogs, measured self-reported stress [35,78]. Binfet [35] described the intervention as AAT whilst the intervention in Ward-Griffin et al. [78] was consistent with AAA. The sessions lasted varying amounts of time from 20 min [35] to up to 90 min (but on average 30 min) [78]. Neither study used a stressor but one study [78] occurred during mid-term exam season. Despite the differences between the two studies, both showed a statistically significant reduction in stress when measured within 24 h of the intervention. This effect was not sustained at the two-week follow-up for the one study that included longer follow-up [35]. Therefore, these studies showed cautious preliminary evidence of a short-term, statistically significant, beneficial effect on stress, using AAIs, with students at Canadian universities. However, with such a small number of studies (with both being at high risk of bias for the outcomes of interest), caution is needed regarding generalisability to other countries and settings.

#### 4.5.3. Secondary Outcomes: Depression, Mood/Affect and Well-Being

The evidence for depression is only based on two studies with different study characteristics and with mixed results [75,77]. One study, with low strength of evidence, showed a non-statistically significant beneficial effect on depression (with a stressor) [77] but we are unable to state if this is the true effect as the study could have been underpowered. The other study showed a reduction in depression scores for both intervention and comparator but by a larger degree for the control (without a stressor) [75]. However, caution is required due to the data’s distribution (skewed) and how the results were provided (mean) personal communication [81].

The evidence for mood is mixed with five studies measuring this outcome [37,72,73,76,78]. A non-statistically significant detrimental effect appeared to occur particularly when a stressor is applied. Of the two studies that showed statistically significant beneficial results without a stressor, this result had not lasted by the 4th session for one study. Where a non-statistical significance was shown, a power calculation was not included. Therefore, the studies may have been underpowered to demonstrate the true effect (which may or may not be similar to the results explored here). Further investigation is required before any conclusions can be made for this outcome.

Three studies measured well-being using various tools (for example, Sense of Belonging in School, Warwick–Edinburgh Mental Well-being Scale (WEMWBS), Satisfaction with Life Scale (SWLS), Subjective Happiness Scale and Medical Outcomes Study Social Support) [35,37,78]. Overall, tentative but mostly beneficial effects were found in varying measures of well-being when measured immediately after or within 24 h of the AAI. Where a non-statistically significant effect was found, the outcomes were measured within 24 h of the intervention during mid-term exam season. Additionally, no power calculation had been included for the non-statistically significant findings, and therefore, distinguishing between true “no effect” or being underpowered was not possible.

## 5. Discussion

### 5.1. Statement of Principal Findings

This systematic review included 11 RCT assessing AAIs on mental health outcomes for students attending higher education in a variety of settings and countries. The evidence suggests that AAIs could provide short-term beneficial results for anxiety in students attending higher education. There is limited evidence for stress, and inconclusive evidence for depression, well-being and mood. These results are from studies at high risk of bias for the outcomes of interest with mostly medium strength of evidence.

### 5.2. Strengths and Weaknesses of the Review

The strengths of this review include a comprehensive search strategy incorporating both grey and published literature. Additionally, independent conduct of screening, data extraction and risk of bias with inclusion of a third reviewer to resolve any discrepancies were incorporated. This reduced the introduction of random and systematic errors [82]. Development of decision rules, where required, increased transparency and rigor. Furthermore, this review combined RCTs, which represent the gold standard study design for evaluating a causal relationship and for measuring an intervention’s effectiveness [83]. A meta-analysis was not appropriate for reasons already described; therefore, vote counting, based on direction of effect and not statistical significance, was employed. Vote counting using statistical significance can be misleading especially in studies where no power calculation was reported and no significance was found [40]. The lack of statistically significant effect may either be due to the study being underpowered or may reflect a true lack of effect [40]. However, when using vote counting on direction of effect without considering statistical significance, the effect seen may be due to chance. Additionally, vote counting methods are unable to provide a precise estimate of the overall effect size.

Due to COVID-19, some resources were inaccessible (list included in Appendix C). Additionally, a pragmatic approach to the Advanced Google search was incorporated using only primary outcomes. Well-being is a particularly broad concept. Proxies that were strongly related to well-being were included but may not have been an exhaustive list. Therefore, some articles that could have met the eligibility criteria may have been missed. A percentage above a minimum for double screening, data extraction and risk of bias assessment was chosen due to resource and time constraints, which represents a limitation. Using the strength of evidence approach, overall weight was influenced mainly by how well the authors described randomisation and/or presence of withdrawals/missing data as all the studies were relevant RCTs where double blinding was difficult. Where not reported, concluding whether these elements simply had not occurred or had occurred but not described due to reasons (such as word-count limits) was impossible, unless clarified by the authors through correspondence. Furthermore, different tools to assess strength of evidence or risk of bias may have produced different findings.

### 5.3. Strengths and Weaknesses of the Studies

The limitations of the primary evidence reviewed included variable levels of descriptive reporting in the included studies, such as participant characteristics, delivery of the intervention and theory of change. For example, the descriptions of the included AAIs varied considerably from a high level of detail to only a single sentence or short paragraph. Therefore, providing conclusions about whether certain types of AAIs were more effective that others or to isolate the “active ingredient” of effective AAIs was not possible. Most of the participants were females, ranging from 57% to 85%, and authors rarely commented on how well the sample population represented the target population. Furthermore, all the participants volunteered to participate and, therefore, may not be representative of the target population [84]. For example, those volunteering (and therefore, self-selecting) may be more motivated and/or have a different mental health status than those who did not [84,85]. Additionally, individuals who are afraid, allergic or have a medical condition precluding participation with animals are unlikely to have volunteered. Therefore, building a comprehensive picture of the type of individual attending higher education, who would benefit from AAIs was challenging. These reporting issues meant that assessing the generalisability of the results was also problematic.

The outcomes reviewed were self-reported and can be subject to unconscious and/or conscious ascertainment bias. With this type of intervention, blinding of participants or individuals delivering the intervention was difficult, if not impossible. This resulted in all the studies having an overall high risk of bias for the outcomes of interest for this systematic review. Additionally, participant expectancy bias could have been introduced, which was highlighted in Williams et al. [79] as 90% of the control group sampled (*n* = 15) stated they thought an interaction with the dog would have reduced their stress prior to an exam. Despite these limitations, capturing how an individual feels after an intervention/comparator is important. To aid corroboration and confidence with self-reported results, triangulation would be beneficial such as with objective measures (for example, physiological outcomes) and/or blinded behavioural observations [86]. Indeed, some of the studies reported physiological outcomes but were beyond the scope of this systematic review, which is a limitation.

Furthermore, for the outcomes where no statistical difference was found, either no power calculation was included, or the study was underpowered. In those studies, no definitive conclusion can be derived about the effectiveness of that particular intervention for that specific outcome as it is not possible to state if the non-statistically significant difference was the true effect or not [87].

### 5.4. Study Meaning: Possible Mechanism and Implications for Policymakers

The Fogg Behaviour Model (FBM) is a theory of change model that could be considered for AAIs and student engagement to improve mental health outcomes [88]. In summary, FBM requires three elements for the intended behaviour of student engagement with AAIs to occur: motivation, ability and triggers [88]. Triggers promote the intended behaviours and may be achieved by advertisement/promotion of the sessions [88]. Motivation for attendance is proposed as the animals’ presence addressing three core motivators: (1) hope of an experience that is likely to be (2) pleasurable and (3) socially acceptable for the majority [88]. Finally, to optimise the students’ ability to participate in this intended behaviour, the model’s six elements of simplicity should be addressed [88]:time (sessions to be short)money (sessions to be cost-neutral for students)physical effort (sessions to be offered in an accessible location)brain cycles (process by which to attend the sessions should be easy)social acceptance (as offered by activities with animals)routine (regular sessions to be offered)

For policymakers considering implementing AAIs in a higher educational setting, a logic model should be developed alongside the intervention to assist in clarifying the active ingredients and causal assumptions [89]. Key stakeholders, such as students from varying backgrounds, staff from student support services and animal/handler teams, should be involved during the design stage. A formative evaluation, including both process and implementation assessments, with a mixed-methods pilot study would be a useful first step [89,90,91]. The feasibility, acceptability and fidelity of the AAI in the target population can therefore be assessed with adaptation, if required. Thereafter, a summative evaluation, using a larger mixed-methods RCT, evaluating effectiveness, with a nested process evaluation, would be recommended [90].

### 5.5. Future Research Recommendations

Further research recommendations are detailed in Box 1. Particularly, the overall reporting quality by authors should be improved with facilitation from journals. For example, authors should provide enough information to allow replication of the interventions or expansion of the existing research [65]. These steps will help identify the effective or ineffective interventions and facilitate further evaluation of the active ingredients. Furthermore, potentially conflicting evidence, albeit not statistically significant, is present for mood which needs further evaluation to ensure that AAIs do not have unintended negative consequences [92]. Additionally, including animal welfare and economic evaluations are important to help secure support and funding from commissioners.

Box 1The recommendations for future research for AAIs.
Use of standardised and internationally recognised definitions when describing AAIsUse of sample sizes that provide adequate powerClarity regarding the randomisation procedure (including description of allocation, and whether concealed allocation occurred) and provision of an adequate description of the participants’ characteristics separated by groupClear reporting of the participants’ flow through the trial with reasons for any missing data for each respective group and at each time-pointUse of explicit comparators to establish the relative effects of the co-interventions (e.g., appropriate attention controls)Adequate descriptions of the interventions implemented to facilitate replicationClear reporting of the outcome measurement procedure (particularly when multiple time-points or stressors are present), including any adaptations made to the scales usedProvision of access to publicly available pre-specified statistical analysis plans by authors, including justification for choice of target differencesClear reporting of adverse events for both humans and animals


## 6. Conclusions

Animal-assisted interventions (AAIs) were considered as potential interventions to help improve mental health and well-being of students in higher education who are willing to and can engage with animals. The pooled evidence suggests that AAIs could provide short-term beneficial results for anxiety, and possibly stress, in this population, known to be at risk of mental health issues. However, caution is required as these results were from studies at high risk of bias for the outcomes of interest for this systematic review with mostly medium strength evidence, and in various cultural settings. Subsequent implementation of AAIs in this setting requires both formative and summative evaluation to measure both the intended and unintended consequences. Furthermore, consideration of alternatives for students unable to participate due to fear of animals or medical contraindications is recommended to prevent widening of health inequalities.

## Figures and Tables

**Figure 1 ijerph-18-10768-f001:**
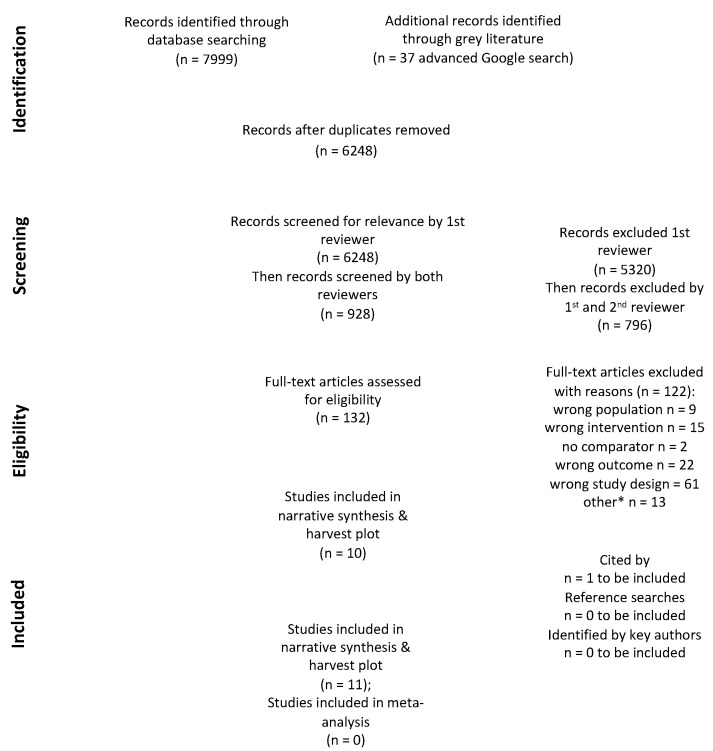
Adapted preferred reporting items for systematic reviews and meta-analysis (PRISMA) diagram adapted from Moher et al. [39]; * other included not available due to COVID-19 (*n* = 2), did not meet criteria for requesting inter-library loan (*n* = 10) & ongoing trial (*n* = 1).

**Figure 2 ijerph-18-10768-f002:**
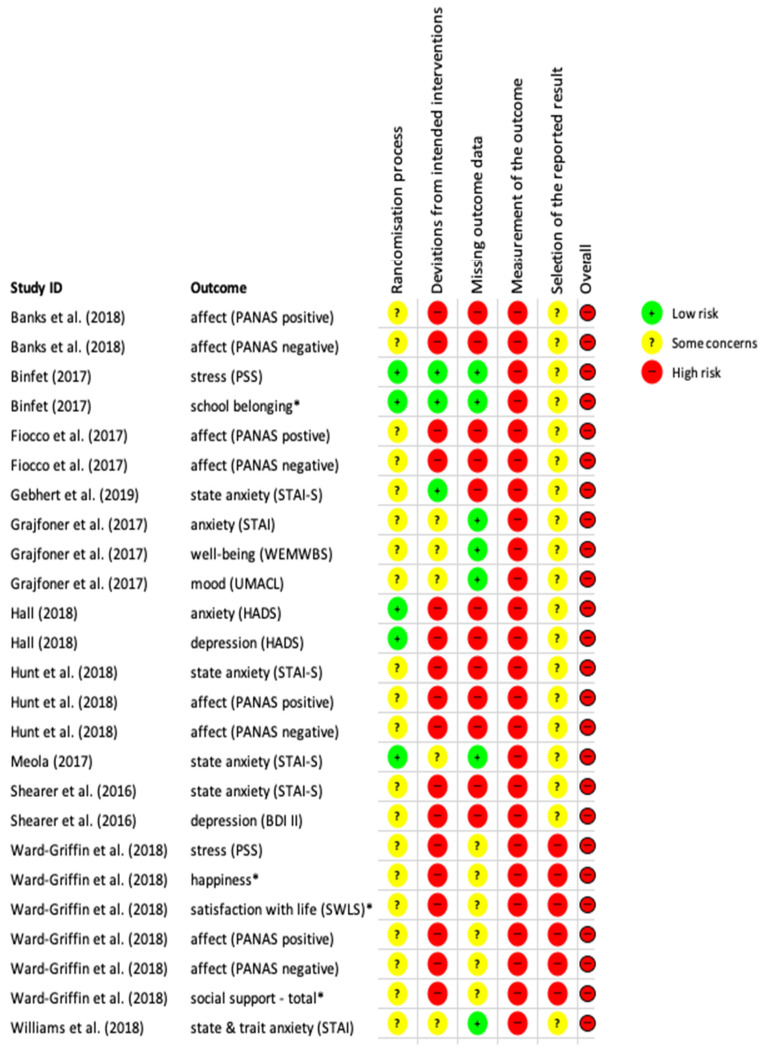
Risk of bias assessment (RoB2 assessment) for all included studies on the outcomes of interest; * considered as a proxy for well-being.

**Figure 3 ijerph-18-10768-f003:**
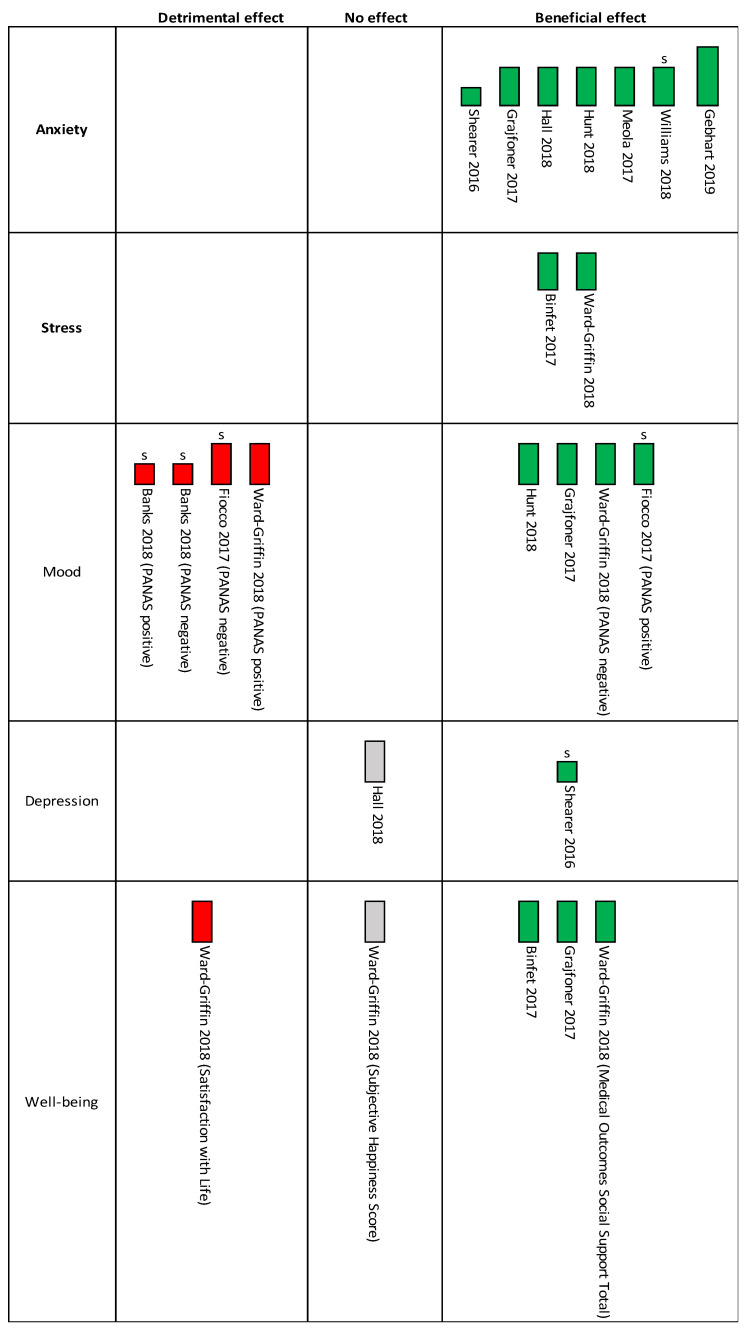
Harvest plot for the primary & secondary outcomes for one comparator & one time-point as described in the methods. s stands for stressor; height of bars relate to overall strength of evidence (tallest = high; shortest = low).

**Table 1 ijerph-18-10768-t001:** Summary of inclusion & exclusion criteria with rationale [7,8,40,41,42,43,44,45,46].

Inclusion	Exclusion
**Population:** Students in higher education (defined as post-secondary education leading to a degree). OR A description of an equivalent/use of terms known to be associated with higher education. Rationale: Definition of higher education and represents a population exposed to significant stress. If stressor present, had to be an aspect of study, training, education or be student-specific. Rationale: student-specific stressor.	**Population:** Students who all have an established diagnosed condition/disorder (such as autism or ADHD). Rationale: could substantially affect the clinical heterogeneity of the populations being compared.
**Intervention:** AAI—particularly AAT and AAA. OR Live animal considered/called a therapy animal (OR animal had training AND assessment or evaluation/certification), a therapeutic goal/aim was identified, and the outcomes of interest were evaluated. Rationale: Key elements of AAI (including AAT and AAA); evaluation of relevant outcomes was required to assess results.	**Intervention:** Not involving a live animal. Rationale: AAIs involve live animals. Participants’ own pets/companion/support/assistance/service animals. Rationale: Likely to represent potential confounders/effect modifiers and not consistent with definitions of AAI, AAT or AAA.
**Comparator:** A comparison group required. Rationale: comparators are required to evaluate intervention’s effectiveness.	**Comparator:** No comparator.
**Outcome:** Psychological using published or established standardised measures: **Primary outcomes:** effect on anxiety and/or stress. **S****econdary outcomes:** effect on depression, mood/affect and well-being. Rationale: Represent important measures of mental health and well-being.	**Outcome:** Physiological. Rationale: Often used as proxy measures for psychological states but not directly related to psychological outcomes. Educational/or academic. Rationale: Focus is on mental health and well-being, not performance.
**Study:** RCT and other types of randomisation. Rationale: RCT represents gold standard for measuring effectiveness.	**Study:** all non-randomised. Rationale: prone to effects of confounding & to ensure feasibility of review due to time/resources constraints.

**Table 2 ijerph-18-10768-t002:** Characteristics of included studies ordered alphabetically with relevant outcome measures (NR = not reported; SD = standard deviation).

First Author, Year & Country	Participants	Intervention			Comparator	Outcomes	
	Characteristics	Theoretical Framework Articulated	Description of Intervention	Type of AAI & Delivery		Tools Used	Time-Points
Banks [72] 2018 USA	University students with some recruited from psychology department (76.8% female) Mean age 20.05 (SD 3.38) Year of study, type of graduate, ethnicity, health status or SES NR	No	Group (free interaction) with as many dogs as wanted (student: dog ratio NR) for 10 min single session during mid-term exam week Various breeds (e.g., Beagle, Golden Retrievers, German Shepherds)	AAA Handler	No-treatment control	PANAS—positive & negative	Pre & post stressor (SART & letter/ pattern comparison)
Binfet [35] 2017 Canada	First-year university undergraduate students taking psychology classes (78% female) Mean age 18.85 (SD 2.65) Ethnicity: 57% Caucasian; 15% Chinese; 9% Mixed-Race Health status or SES NR	Yes	Group (free interaction) 3–4 student: 1 therapy dog/handler for 20 min single session Various breeds including pure-bred & mixed (which breeds NR)	AAT * Handler	No-treatment control (“business as usual” control = studying)	PSS Sense of Belonging in School	Pre, post, & then follow-up after 2 weeks
Fiocco [73] 2017 Canada	Undergraduate university students (77.1% female) Mean age 21.02 (SD 5.5) Ethnicity: 37.7% Caucasian; 8.2% Black/African American; 54.1% Other Year of study, health status or SES NR	Partial	Individual (free interaction as long as participant remained seated) with a dog for 10 min single session Various breeds of different ages & sizes/breed (e.g., Irish Setter, Schnoodle, Greyhound, King Charles Spaniel)	AAA Unclear if handler present	No-treatment control (sitting for 10 min)	PANAS—positive & negative	Pre, post AAA & then after stressor (PASAT)
Gebhart [74] 2019 Austria	First-year students at nursing school (77% female) Median age 20 (IQR 19–22) Type of graduate not clearly specified; health status, ethnicity or SES NR	Partial	Group interaction (structured with different tasks, playing & interacting with dogs) student: dog/handler ratio implied 3 students: 1 therapy dogs/handler for 45–60 min for 3 sessions (time interval between sessions NR)	AAT Handler	No-treatment control unstructured free hour; music therapy (body percussion) & mandala painting	STAI-S	Pre & post normal day Pre & post exam day
Grajfoner [37] 2017 Scotland	University students (64.4.% female) Mean age 21.6 (SD 3.4) Year of study, type of graduate, health status, ethnicity or SES NR	Yes	Group (free interaction) ~6 students: 1 dog/handler ratio for 20 min single session Various breeds (e.g., Labrador, Lhasa Apso, Golden Retriever)	AAA Handler	Handler only (HO) & dog only (DO) (handler present; no interaction)	STAI WEMWBS UMACL	Pre & post
Hall [75] 2018 USA	Level 2 community college associate degree nursing programme Gender, age, year of study, type of graduate or health status, ethnicity or SES NR (only demographics of the course)	Partial	Mix of group or individual session (free interaction) with numerous opportunities to interact with dog. Dog on campus minimum twice a week, visited students at various locations & on exam days for 30 min pre-exams. Intensity, length & frequency of sessions NR Standard Poodle	AAA Handler (unclear if always present)	No-treatment control	HADs—anxiety & depression	Pre & post (which is after 15–16 weeks from start of intervention)
Hunt [76] 2018 USA	Undergraduate students enrolled in psychology courses (74% female) Mean age 19.3 (SD NR) & ≥18 yrs old Ethnicity: 52% non-Hispanic White; 27% Asian/Asian-American; 9% Hispanic/Latino; 5% Black/African American, 4% Multiracial, 2% Indian; 1% Arab Year of study, health status or SES NR	Partial	Group interaction (free interaction) with a dog plus interactive games, icebreakers & snacks student: dog ratio not clearly stated but implied 12–14 students: 1 dog of unclear length for once/week for 4 sessions Golden Retriever	AAA Unclear if handler present	No-treatment control; mindfulness training alone; yoga alone; or mindfulness training with yoga	STAI-S PANAS—positive & negative	Pre & post every session & once 1–3 weeks after completion of AAA with stressor (WAIS-IV IQ test)
Meola [80] 2017 USA	University students enrolled on accredited counselling program (85% female) Mean age 30.8 (SD NR) Ethnicity: 85.7% Caucasian; 9.5% African American Year of study, type of graduate (some Masters & PhD but NR for all), health status, or SES NR	Yes	Individual (structured & tailored) equine-assisted learning supervision (EALS) session with a horse with 3 different activities: for 1-h single session	AAT/AAE Handler (who was also an instructor/ facilitator)	No-treatment control	STAI-S	Pre & then post up to one month after intervention
Shearer [77] 2016 USA	Undergraduate university students in psychology courses (57% female) Ethnicity: 43% Asian; 41% Caucasian; 7% Hispanic; 3% African American; 3% Other; 1% Native American; 1% Pacific Islander; 1% unidentified. Year of study, health status, age or SES NR	Partial	Group session (free interaction) with a dog plus games & snacks students: dog ratio not clearly specified but implied 12–13 students: 1 dog for 1 h/week for 4 weeks	AAA Facilitator	No-treatment control (added in 2nd phase) or mindfulness meditation	STAI-S BDI II	Pre & then post each session for 4 weeks & then once 1–2 weeks after completion of AAA with stressor (WAIS-IV)
Ward-Griffin [78] 2018 Canada	University students enrolled in introductory psychology classes (78% female) Mean age 19.4 (SD 3.73) 45.5% first-year students; others NR Type of graduate, health status, ethnicity or SES NR	Partial	Group session (free interaction) with dogs. student to dog ratio not clearly specified with 7–12 dogs present with handlers for up to 90 min single session during mid-term exam season (on average participants spent 30 min in the space)	AAA Handler	Wait-list control	PSS PANAS—positive & negative SWLS Subjective Happiness Scale Medical Outcomes Study Social Support Scale	Pre & within 24 h of AAA
Williams [79] 2018 USA	University graduate students in pharmacy/physical therapy (63.2% female) Mean age 24.42 (SD NR) Year of study, health status, ethnicity, or SES NR	Partial	Not clearly specified but inferred as individual free interaction (as long as no active running & playing) with a dog for 12 min single session prior to exam	AAA Handler	No-treatment control (“business as usual” control = quiet time studying)	STAI-S & T	Pre & post (but before an exam)

* as described by the authors; limited information to provide an objective assessment of type of AAI. Abbreviations used: AAA: Animal-Assisted Activity; AAE: Animal-Assisted Education; AAI: Animal-Assisted Intervention; AAT: Animal-Assisted Therapy; BDI-II: Beck Depression Inventory II; HADS: Hospital Anxiety and Depression Scale; IQR: interquartile range; NR: not reported; PANAS: Positive and Negative Affect Schedule; PASAT: Paced Auditory Serial Addition Task; PSS: Perceived Stress Scale; SART: Sustained Attention to Response Task; SES: socioeconomic status; SD: standard deviation; STAI: State-Trait Anxiety Inventory; STAI-S: State-Trait Anxiety Inventory state anxiety; STAI-T: State-Trait Anxiety Inventory trait anxiety; SWLS: Satisfaction with Life Score; WAIS-IV IQ: Wechsler Adult Intelligence Scale-IV.

**Table 3 ijerph-18-10768-t003:** The strength of evidence for the included studies in accordance with the ’Weight of Evidence’ approach [70].

Study	Trustworthiness (A)	Appropriateness (B)	Relevance (C)	Overall Weight (D)
Banks et al. [72]	Low	High	High	Low
Binfet [35]	Medium	High	High	Medium
Fiocco et al. [73]	Medium	High	High	Medium
Gebhart et al. [74]	High	High	High	High
Grajfoner et al. [37]	Medium	High	High	Medium
Hall [75]	Medium	High	High	Medium
Hunt et al. [76]	Medium	High	High	Medium
Meola [80]	Medium	High	High	Medium
Shearer et al. [77]	Low	High	High	Low
Ward-Griffin et al. [78]	Medium	High	High	Medium
Williams et al. [79]	Medium	High	High	Medium

## Data Availability

Not applicable.

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
