# Peer review of "Animal-Assisted Interventions for the Improvement of Mental Health Outcomes in Higher Education Students: A Systematic Review of Randomised Controlled Trials"

_ijerph, 2021, doi:10.3390/ijerph182010768_

Round 1
Reviewer 1 Report
This manuscript provides a systematic review of the literature around animal-assisted activities for university students, with outcomes especially relating to stress and anxiety. This ms is well-written and could be a useful contribution to the literature, but I have one major concern:
The authors argue that there is a high risk of bias in all studies due to self-report measures, but they intentionally excluded studies that made use of objective physiological measures. They note this in the discussion as a limitation, but that does not change their overall assertion that the risk of bias is high. This seems frankly unfair to the studies that did include a physiological measure. The authors should not intentionally exclude them and then claim that the quality of the evidence is poor, when their own inclusion criteria all but guaranteed that.
Furthermore, for the self-report measures, it is possible, as the authors state, to introduce expectancy bias. But surely a validated measure should be considered to be of higher quality than a non-validated one. In this field, since it is impossible to blind participants to the condition, thus making any study at high risk of bias, it may be necessary to alter the criteria slightly, so that even though the participants cannot be fully blinded to the condition, they could be potentially recruited without being fully informed of the study's aims. That could be considered more heavily than just blind/non-blind for this field, which should still fit within Cochrane's guidelines. That is, in theory, it should be possible for a participant to know that they are part of a study where there may be an animal, but not know that the purpose of the intervention is to make them feel more relaxed. Is it possible to consider the 11 included studies on that basis? It could generate a bit more variability in the quality of the studies that way, which would make the entire review more useful.
Other comments
Strong intro - well done
L153 - could the authors please confirm that the Google search was intended to locate grey literature? Or was Google Scholar used to search scientific literature?
L164 - suggest moving Table 1 just below this para, right after where it is first mentioned in text.
L202 - please provide an example of a stressor
L237 - the way it's written isn't entirely clear to me: were the authors noting those examples (e.g., date of birth, student number) as examples of allocation that WAS or WAS NOT objectively randomised?
L261 - how often did the authors not agree?
L282 - as above re: disagreement frequency
L365 - as a minor semantic comment, I suggest modifying the term 'used', as the field is attempting to move towards less-servile language when referring to working animals. Perhaps 'employ' would be better for this
Table 2 - in several instances, the authors refer to 'therapy dog' in studies that were using AAA, rather than AAT. It is inappropriate to refer to dogs in AAA as therapy dogs, so suggest changing to just 'dogs' throughout.
Table 2 again - Please provide a footnote at the end of table with all of the measures including their abbreviations and their full titles. This is done in Appendix D but it should be made available as part of this table. Tables and Figures should be able to stand alone so that readers can read just the tables/figures without also needing to read the main text to make sense of them.
Table 2 again - Is it possible to elaborate slightly on the theoretical framework used in the studies where one is mentioned? Maybe the name of the theory?
L442 - why the scare quotes around 'weight' in this para?
L469 - same issue re: therapy dogs in AAA instead of AAT
L498 - suggest moving Fig 3 just below here, immediately after it is referred to in text.
L460/500 - why do only 9 of the 11 studies have anxiety or stress if those are the primary outcome measures? Should the remaining two even be included at all if they do not assess the primary outcome measures?
Appendix D large table should be included in the main text. It is an important summary of all the results and should not be relegated to an appendix. Appendices should be reserved for information that is potentially useful to the handful of people who might want to replicate the review, but is not relevant to the story in the main text. This table is relevant to the story.
Appendix D large table has same issues as Table 2 re: 'therapy dog' in AAA. Also, please explain what HO and DO represent in the Grajfoner study column 2.
Author Response
Author's reply to Reviewer 1:
Point 1: This manuscript provides a systematic review of the literature around animal-assisted activities for university students, with outcomes especially relating to stress and anxiety. This ms is well-written and could be a useful contribution to the literature, but I have one major concern:
The authors argue that there is a high risk of bias in all studies due to self-report measures, but they intentionally excluded studies that made use of objective physiological measures. They note this in the discussion as a limitation, but that does not change their overall assertion that the risk of bias is high. This seems frankly unfair to the studies that did include a physiological measure. The authors should not intentionally exclude them and then claim that the quality of the evidence is poor, when their own inclusion criteria all but guaranteed that.
Furthermore, for the self-report measures, it is possible, as the authors state, to introduce expectancy bias. But surely a validated measure should be considered to be of higher quality than a non-validated one. In this field, since it is impossible to blind participants to the condition, thus making any study at high risk of bias, it may be necessary to alter the criteria slightly, so that even though the participants cannot be fully blinded to the condition, they could be potentially recruited without being fully informed of the study's aims. That could be considered more heavily than just blind/non-blind for this field, which should still fit within Cochrane's guidelines. That is, in theory, it should be possible for a participant to know that they are part of a study where there may be an animal, but not know that the purpose of the intervention is to make them feel more relaxed. Is it possible to consider the 11 included studies on that basis? It could generate a bit more variability in the quality of the studies that way, which would make the entire review more useful.
Reply to point 1: Thank you for your comments. The RoB 2.0-Revised tool, a gold standard tool for quality appraising RCTs produced by the Cochrane Collaboration, was used to assess risk of bias. I appreciate that using another tool may have produced different results, so this has been highlighted in the discussion (L 1373-1375). However, we are likely to have encountered similar issues with alternative tools, for example when assessing blinding. For example, even if the participants were blinded to the hypothesis, it is likely that the participants might be able to deduce the hypothesis, even if blinded from it, due to the types of measures that the participants were being assessed on. Additionally, deviation from any established and validated tools may not produce reproducible results.
Risk of bias assessments were completed for each specific outcome of interest for this systematic review. Therefore, the risk of bias assessments comment only on the specific associations between the intervention and these specific outcomes of interest rather than an overall assessment of the studies themselves. Therefore, I have adjusted the wording in the manuscript to specify that the risk of bias assessments are for those outcomes of interest for this systematic review (L390-391, 961, 967, 980, 1176, 1247, 1282, 1381, 1467). The studies which included physiological measures are likely to have been more robust, however, it was not feasible to include this evidence into the current review as well. Further research could explore the association between animal-assisted interventions and physiological measures of anxiety, stress and wellbeing and to triangulate the evidence from subjective measures. This has been included as a limitation in the discussion.
Point 2: Strong intro - well done
Reply to point 2: Thank you
Point 3: L153 - could the authors please confirm that the Google search was intended to locate grey literature? Or was Google Scholar used to search scientific literature?
Reply to point 3: It was an advanced Google search and not a search using Google Scholar.
Point 4: L164 - suggest moving Table 1 just below this para, right after where it is first mentioned in text.
Reply to point 4: Thank you for the suggestion, this has been changed (now L 219).
Point 5: L202 - please provide an example of a stressor
Reply to point 5: Two examples of a stressor have been included (L 265/266).
Point 6: L237 - the way it's written isn't entirely clear to me: were the authors noting those examples (e.g., date of birth, student number) as examples of allocation that WAS or WAS NOT objectively randomised?
Reply to point 6: These are examples of not being objectively randomised, so the sentence has been amended to make it clearer (L 298/299).
Point 7: L261 - how often did the authors not agree?
Reply to point 7: 11 articles were reviewed by the third reviewer at the study selection phase.
Point 8: L282 - as above re: disagreement frequency
Reply to point 8: 0 for the data extraction phase.
Point 9: L365 - as a minor semantic comment, I suggest modifying the term 'used', as the field is attempting to move towards less-servile language when referring to working animals. Perhaps 'employ' would be better for this
Reply to point 9: Thank you, the word used when involving the animals has been changed to employed as suggested (L 571, 574, 993).
Point 10: Table 2 - in several instances, the authors refer to 'therapy dog' in studies that were using AAA, rather than AAT. It is inappropriate to refer to dogs in AAA as therapy dogs, so suggest changing to just 'dogs' throughout.
Reply to point 10: Thank you, these have been changed unless it is clear it was AAT.
Point 11: Table 2 again - Please provide a footnote at the end of table with all of the measures including their abbreviations and their full titles. This is done in Appendix D but it should be made available as part of this table. Tables and Figures should be able to stand alone so that readers can read just the tables/figures without also needing to read the main text to make sense of them.
Reply to point 11: To aid with the understanding of the abbreviations, a list has been provided at the end of the article (L 1496) as well as below this Table (L 947-950).
Point 12: Table 2 again - Is it possible to elaborate slightly on the theoretical framework used in the studies where one is mentioned? Maybe the name of the theory?
Reply to point 12: Yes, these were removed due to space. For a study to be classed as clearly stating the theoretical framework that underpinned the included intervention, one of the following criteria was required:
- mechanism of action was stated and directly linked back to the interventions' development before implementation; or
- proposal was offered for the intervention's mechanism of action on the outcomes before the intervention was implemented; or
- mechanism of action was stated with a pre-specified assessment to distinguish the different co-interventions’ relative effects.
This information has been included in Appendix D.
Point 13: L442 - why the scare quotes around 'weight' in this para?
Reply to point 13: These quotation marks have been removed.
Point 14: L469 - same issue re: therapy dogs in AAA instead of AAT
Reply to point 14: Thank you, this has been changed.
Point 15: L498 - suggest moving Fig 3 just below here, immediately after it is referred to in text.
Reply to point 15: Thank you, this suggestion has been actioned (L1223).
Point 16: L460/500 - why do only 9 of the 11 studies have anxiety or stress if those are the primary outcome measures? Should the remaining two even be included at all if they do not assess the primary outcome measures?
Reply to point 16: Both primary and secondary outcomes were considered when reviewing the articles for inclusion. Therefore, it was not a prerequisite of this review that all of the included studies had to have measured this review's primary outcomes, or to have designated those outcomes as ‘primary’ in the original studies.
Point 17: Appendix D large table should be included in the main text. It is an important summary of all the results and should not be relegated to an appendix. Appendices should be reserved for information that is potentially useful to the handful of people who might want to replicate the review, but is not relevant to the story in the main text. This table is relevant to the story.
Reply to point 17: I would be happy to include as part of the paper proper, but I have concerns about the effect this would have on the length of the paper and word count. Can the editor be consulted about the best course of action please?
Point 18: Appendix D large table has same issues as Table 2 re: 'therapy dog' in AAA. Also, please explain what HO and DO represent in the Grajfoner study column 2.
Reply to point 18: Thank you, the use of therapy dog has been amended unless the intervention was AAT. DO and HO have been clarified - handler only HO; dog only DO.
Reviewer 2 Report
A very extensive and carefully developed systematic review, with many references and careful analysis, identifying modest short-term benefits of animal assisted interventions. Although a very long article, it is justified by being generally well written with thoughtful organization, and good attention to details. Could be a model of how to conduct a systematic review of a complex medical intervention.
Specific suggestions:
There are many acronyms, perhaps of table of them would be useful.
Lines 58-60, the study reviews articles that address AAI, AAT, AAF, AAA, and AAC. Perhaps just a line to basically describe these would help the read not as steeped in the subject. AAC may be new to even people familiar with the other areas of study.
Line 78, list suggested reasons why animals bestow health effects on people using only one reference. The authors may also consider:
Beck AM. The biology of the human–animal bond. Animal Frontiers. 2014 Jul 1;4(3):32-6.
Line 144 Peer-reviewed strategy, nice touch.
Line 205, word “comparator” not typical but good discussion.
Line 420, should there not be some comment on the observation that there were no adverse events with all these studies?
Lines 432-436, good discussion about bias, but perhaps you could suggest improvements?
Lines 553-556, good assessment of the articles strengths.
Line 652, Box 1 is good.
Lines 656-658, Conclusions are good and fair.
Tables are large, busy, and wordy but good information Consider some streamlining.
Figures a bit over-produced but will be appreciated.
A useful contribution to the HAI field.
Author Response
Authors' reply to Reviewer 2's comments:
Point 1: A very extensive and carefully developed systematic review, with many references and careful analysis, identifying modest short-term benefits of animal assisted interventions. Although a very long article, it is justified by being generally well written with thoughtful organization, and good attention to details. Could be a model of how to conduct a systematic review of a complex medical intervention.
Reply to point 1: Thank you for your time and comments, they are much appreciated.
Specific suggestions:
Point 2: There are many acronyms, perhaps of table of them would be useful.
Reply to point 2: A list has been produced and included after the conflicts of interest (L 1496).
Point 3: Lines 58-60, the study reviews articles that address AAI, AAT, AAF, AAA, and AAC. Perhaps just a line to basically describe these would help the read not as steeped in the subject. AAC may be new to even people familiar with the other areas of study.
Reply to point 3: Have included brief information in the introduction explaining the different terms for AAT, AAA, AAE and AAC (L 63-72).
Point 4: Line 78, list suggested reasons why animals bestow health effects on people using only one reference. The authors may also consider:
Beck AM. The biology of the human–animal bond. Animal Frontiers. 2014 Jul 1;4(3):32-6.
Reply to point 4: I have changed the formatting of this paragraph to make it clearer as it is not only the theories discussed by Crossman et al that are included but also Biophilia Theory and psychosocial model. The suggested reference has also been included - many thanks (L 99).
Point 5: Line 144 Peer-reviewed strategy, nice touch.
Reply to point 5: Thank you
Point 6: Line 205, word “comparator” not typical but good discussion.
Reply to point 6: Thank you.
Point 7: Line 420, should there not be some comment on the observation that there were no adverse events with all these studies?
Reply to point 7: None of the papers reported adverse events so it is difficult to comment further on this apart from stating it. I have included in the discussion the importance of reporting this especially for commissioners considering implementing AAIs (L 925, 926 and Box 1).
Point 8: Lines 432-436, good discussion about bias, but perhaps you could suggest improvements?
Reply to point 8: It has now been acknowledged that using a different risk of bias tool alongside the strength of evidence tool might possibly have produced different findings (L 1367).
Point 9: Lines 553-556, good assessment of the articles' strengths.
Reply to point 9: Thank you.
Point 10: Line 652, Box 1 is good.
Reply to point 10: Thank you.
Point 11: Lines 656-658, Conclusions are good and fair.
Reply to point 11: Thank you.
Point 12: Tables are large, busy, and wordy but good information. Consider some streamlining.
Reply to point 12: Thank you. The tables have been streamlined from the Masters' dissertation and adapted to ensure as succinct as possible. A balance between both reviewers has been attempted, can we defer any further changes to the editor please?
Point 13: Figures a bit over-produced but will be appreciated.
Reply to point 13: Thank you.
Point 14: A useful contribution to the HAI field.
Reply to point 14: Thank you.
Round 2
Reviewer 1 Report
This ms has been much improved. I am happy to accept it for publication. Well done.